# Clinical and Histological Effects of Partial Blood Flow Impairment in Vascularized Lymph Node Transfer

**DOI:** 10.3390/jcm11144052

**Published:** 2022-07-13

**Authors:** Shinsuke Akita, Yuzuru Ikehara, Minami Arai, Hideki Tokumoto, Yoshihisa Yamaji, Kazuhiko Azuma, Yoshitaka Kubota, Hideaki Haneishi, Motoko Y. Kimura, Nobuyuki Mitsukawa

**Affiliations:** 1Department of Plastic, Reconstructive and Aesthetic Surgery, Graduate School of Medicine, Chiba University, Chiba 260-8670, Japan; t.minami.a@gmail.com (M.A.); funkmastery@gmail.com (Y.Y.); 2m2hy4@gmail.com (Y.K.); nmitsu@air.linkclub.or.jp (N.M.); 2Department of Pathology, Graduate School of Medicine, Chiba University, Chiba 260-8670, Japan; yuzuru-ikehara@chiba-u.jp (Y.I.); kazuma@office.chiba-u.jp (K.A.); 3Department of Plastic and Reconstructive Surgery, Chiba Cancer Center Hospital, Chiba 260-0801, Japan; tokumoto0414@yahoo.co.jp; 4Center for Frontier Medical Engineering, Chiba University, Chiba 263-8522, Japan; haneishi@faculty.chiba-u.jp; 5Department of Experimental Immunology, Graduate School of Medicine, Chiba University, Chiba 260-8670, Japan; kimuramo@chiba-u.jp

**Keywords:** vascularized lymph node transfer, lymphedema, lymph node, ischemia, congestion

## Abstract

Regarding vascularized lymph node transfer (VLNT) for lymphedema, partial blood flow impairment in transferred lymph node (LN) flaps may adversely affect the therapeutic results. We investigated the clinical and histological effects of partial blood flow impairment in LN flaps. In upper extremity lymphedema cases, based on ultrasonographic examination at 2 weeks after VLNT, we compared the treatment results depending on whether the postoperative blood flow in transferred LNs was good (Group G) or poor (Group P). Novel partial ischemia and congestion of LN flap mouse models were developed to determine their histological features. In 42 cases, significant differences were observed between Group G (*n* = 37) and Group P (*n* = 5) based on the amount of volume reduction (136.7 ± 91.7 mL and 55.4 ± 60.4 mL, respectively; *p* = 0.04) and lymph flow recanalization rate in indocyanine green fluorescent lymphography (67.6% and 0%, respectively; *p* = 0.0007). In mouse models, thrombi formation in the marginal sinus and numerous Myl9/12-positive immunocompetent cells in follicles were observed in congested LNs. Blood flow maintenance in the transferred LNs is an essential factor influencing the therapeutic effect of VLNT. Postoperatively, surgeons should closely monitor blood flow in the transferred LNs, particularly in cases of congestion.

## 1. Introduction

Vascularized lymph node transfer (VLNT) has been reported to be an effective treatment option for lymph flow reconstruction in patients with lymphedema [1,2,3,4,5,6,7,8]. In previous clinical studies on VLNT for lymphedema, cases with a high therapeutic effect have been reported, although some therapeutic effects were inferior despite the absence of symptomatic complications in the postoperative acute phase [9,10]. As there are cases with insufficient blood flow in the transferred LNs after surgery despite skin paddle survival and no clinical complications in the acute phase, we emphasize the importance of postoperative ultrasound monitoring of the transplanted lymph nodes (LNs) to confirm the maintenance of blood flow and bilayer structure [11]. In this study, partial blood impairment, regardless of arterial blood flow, venous blood flow, or both arterial and venous blood flows, was defined as the condition in which the blood flow of the transferred LN was not confirmed even though the skin paddle survived. In the literature, the adverse effect of partial blood flow impairment in the transferred flap on long-term treatment outcomes has not been proven. Moreover, whether insufficient arterial or insufficient venous blood flow is a more serious problem remains unclear.

In animal models of vascularized, single-venous, and non-vascularized LN transfers, survival rates, as well as lymphatic recanalization with the recipient area, have been reported [12,13,14,15,16]. A study reported that the structure of inguinal LNs was destroyed in a blood vessel occlusion rat model [15]. However, it is unclear how LNs are histologically degenerated under a partially impaired blood flow to the LN flap.

This study aimed to determine the clinical and histological effects of partial blood flow impairment on the transferred LNs after VLNT. The differences in treatment outcomes were compared depending on whether blood flow in postoperative transplanted LNs can be detected or not in patients with upper extremity lymphedema (UEL) who underwent VLNT. In addition, to determine the histological changes, a mouse model of a groin LN flap with partial blood flow impairment was developed.

## 2. Materials and Methods

### 2.1. Clinical Study

#### 2.1.1. Patients

This study was conducted at the Chiba University Hospital (Chiba, Japan). The study protocol was approved by the Institutional Review Board and Ethics Committee (No. 4202). Patients with UEL who underwent VLNT from April 2013 to December 2020 were included in this study.

#### 2.1.2. Surgical Strategy

Surgical treatment was indicated for patients who continued complex physical therapy on an outpatient basis for ≥3 months under the guidance of specialized lymphedema therapists. It reached a plateau of volume changes in the upper extremities. VLNT was indicated for patients with International Society of Lymphology (ISL) stage II disease who wanted a simultaneous breast reconstruction, patients with ISL stage III disease, or patients who had previously undergone lymphaticovenular anastomosis but had unimproved lymphedema.

Before surgery, indocyanine green (ICG) fluorescent lymphography was performed by injecting 0.2 mL of 0.5 mg/mL ICG solution subcutaneously at a 5 cm distal point on the upper arm from the axilla to determine whether a linear pattern of lymph flow from the upper arm into the axilla could be observed or not [17].

In our case series with UEL, after completely resecting the scar tissues, a flap was placed to bridge the soft tissues between the upper arm and trunk [18]. For patients who needed a simultaneous breast reconstruction and VLNT, a combined flap of the deep inferior epigastric artery perforator (DIEP) flap and LN flap fed by the superficial circumflex iliac artery (SCIA) was used [19]. The internal mammary artery and vein were used as recipient vessels for DIEP flaps. The branches of the subscapular artery and vein were used as recipient vessels of the SCIA and vein. Thus, vascular anastomoses of arteries and veins directly connected to the transplanted LNs were performed in all cases. In patients who did not wish to undergo breast reconstruction, those who already underwent breast reconstruction, or those who needed a partial breast reconstruction after breast-conserving surgery, VLNT based on the SCIA or thoracodorsal dorsal artery (TDA) from the contralateral side was performed [20].

The largest LN among those included in the flap was identified by preoperative ultrasonography, and this LN was used to monitor the changes in the morphology of LNs after surgery in postoperative ultrasonography.

#### 2.1.3. Postoperative Course

Compression therapy for lymphedema was restarted immediately after the surgery. The vascular anastomosis site was marked with a pen on the skin and was covered with a clear film, and compression of the site was prohibited for the first 10 days. The postoperative anastomotic thrombus was monitored using continuous tissue oxygen saturation measurements of the skin paddle and periodic color Doppler sonography of the anastomosed vessels for 72 h after surgery [21].

At 2 weeks after surgery, when the risk of total flap failure was considered minimal, ultrasonography of the transferred LNs was performed to determine whether it was possible to examine their blood flow [22,23]. Based on the presence or absence of blood flow signals in LNs at 2 weeks after surgery, the patients were classified as those with good (Group G) and poor (Group P) blood flows. The changes in estimated limb volumes, ultrasonographic findings, and ICG lymphography findings at 6 months after surgery were compared between the two groups. The limb volume was calculated using the circumference of the four points and the distance between the measurement sites [17]. The difference between the preoperative and postoperative limb volumes was defined as the amount of volume reduction achieved by the treatment. ICG fluorescent lymphangiography was performed by injecting an ICG solution at a 5 cm distal point on the upper arm from the distal end of the transferred tissue to determine whether lymphatic flow directing to the transferred LNs was observed. Ultrasonography was performed to observe the transferred LNs and determine whether their blood flow could be detected [11].

### 2.2. Animal Study

#### 2.2.1. Experimental Animal Models 

All research protocols, including the procedures and animal care, were approved by the Institutional Animal Care and Use Committee of Chiba University (Chiba, Japan, approval ID: 2-209). All methods were performed following the relevant guidelines and regulations.

#### 2.2.2. Murine Model of Bilateral Pedicled Abdominal Flaps with Groin LNs

We used 10–12-week-old male C57 black 6 mice purchased from Oriental Yeast Co., Ltd. (Tokyo, Japan), and we developed a mouse model of bilateral pedicled abdominal flaps with groin LNs.

The abdominal flap, including the subcutaneous fat, and bilateral groin LNs based on bilateral abdominal arteries and veins were elevated [24]. The skin paddle design reported by Matsumoto et al. in rats was modified and used in mice to observe their immunostaining characteristics [25]. The flaps that they developed were characterized by the creation of partial ischemic or congestive flaps by elevating abdominal flaps fed by right and left abdominal vessels and ligating the artery or vein on one side [25]. This model was used in the current study to simulate the disease state that caused the LN to have impaired blood flow as part of the partial blood flow impairment of the flap in the clinical case. The skin paddle was U-shaped and included bilateral abdominal wall arteries and veins as vascular pedicles (Figure 1). The flap width was set at 10 mm. The flap was elevated under a microscope, attaching inguinal subcutaneous fat to the lateral side of the skin paddle. Bilateral inguinal LNs and their feeding vessels were included in the flap (Figure 2). After elevating the bilateral pedicled flap, the left artery was ligated to create a unilateral ischemia flap model, or the left vein was ligated to create a unilateral congestion model. Five control flaps without any ligation, five ischemic flaps, and five congestive flaps were prepared.

To maintain a constant postoperative skin tension, prevent peripheral angiogenesis, and accurately measure the extent of the flap survival area, a metal frame was created using a computer-assisted design and a 3D printer; the flap was sewn into the metal frame (Figure 2). The flap and metal frame were coated with polyurethane film to prevent angiogenesis from the abdominal wall. The survival area of the control, ischemic, and congestive flap models at 72 h after surgery was measured using image analysis software (ImageJ; National Institutes of Health, Bethesda, Md, USA) to verify whether the flap models produced the intended partial necrosis with reproducibility. The flap survival rate was calculated using the following formula: flap survival rate = (survived flap area/total flap area) × 100 (percent) [25]. To confirm whether each partial ischemia and partial congestion model produced the characteristic pathological changes in the flap survival area, histological differences in the survival areas of the skin paddles were compared among the control, ischemic, and congestive flaps using hematoxylin and eosin (HE) staining.

#### 2.2.3. Pathologic Features of Ischemic and Congestive LNs

Histopathological specimens of LNs and surrounding tissues were obtained 72 h after flap elevation. The pathological features of the control, ischemic, and congestive LNs were compared. Immunohistochemical staining with HE, Masson’s trichrome, Elastica van Gieson, phosphotungstic acid hematoxylin (PTAH), and myosin light chains 9, 12a, and 12b (Myl9/12) (anti-Myl9/12 Ab, F6 Abwiz bio) was carried out. Myl9/12 immunostaining was performed to observe immunocompetent cell migration with thrombus formation and platelet activation [26,27].

### 2.3. Statistical Analysis

JMP version 13 software (SAS Institute Inc., Cary, NC, USA) was used for all statistical analyses. During comparisons between the two groups in the clinical study, the *t*-test was used to compare continuous variables, and the chi-square test was used to compare frequencies. During the comparison of continuous variables among the three groups in the animal study, multiple tests were performed; the one-way analysis of variance (ANOVA) and the Tukey–Kramer method were used. A *p*-value of <0.05 was considered statistically significant.

## 3. Results

### 3.1. Clinical Study

Forty-two patients with UEL following breast cancer who underwent VLNT were analyzed. Basic patient information and clinical results are presented in Table 1. Twenty-two patients underwent simultaneous DIEP flap breast reconstruction, and their VLNT was based on SCIA. Of the remaining 20 patients who did not undergo simultaneous breast reconstruction, 12 were based on SCIA, whereas 8 were based on contralateral TDA.

No acute phase complications of the flap, such as emergency operation, total necrosis, or partial necrosis, were observed in any cases. In an ultrasonographic study at 2 weeks after the operation, blood flow signals could be detected in the fat layer of the transferred flap in all cases. However, the blood flow in the transferred LNs was clearly identified in 37 patients, and they were classified as Group G; this was not clearly identified in the other 5 patients who were classified as Group P. At 6 months after the surgery, the amount of volume reduction in Group G (142.9 ± 89.4 mL) was significantly larger than that in Group P (62.1 ± 55.0 mL; *p* = 0.03). Ultrasonography at 6 months after the surgery demonstrated that blood flow in LNs was maintained in all patients in Group G. In three cases in Group P, an LN-like structure was identified, but its internal blood flow was not identified. In the other two cases in Group P, LN-like structures could not be identified.

In ICG fluorescent lymphography at 6 months after VLNT, a linear pattern toward the direction of the transferred LNs was observed in 25 patients (67.6%) in Group G; of these, the LN was also fluoresced in 12 cases. In Group P, neither a linear pattern nor fluorescence of the LNs was observed in any cases. There was a significant difference between the groups in terms of frequency (*p* = 0.007, Table 1).

### 3.2. Case Report

A 72-year-old female suffered from right UEL with an accompanying radiation ulcer on the right chest and chronic wound on the forearm after a mastectomy and radiotherapy. She had already undergone conservative treatments under the guidance of specialized lymphedema therapists for years and chest wall reconstruction with a pedicled abdominal flap for a previous radiation ulcer. Although the upper extremity ulcer epithelialized after continuous conservative treatment, the UEL and radiation ulcer on the chest remained (Figure 3A). She underwent VLNT based on contralateral TDA. The recipient’s vessels were the internal thoracic artery and vein. The blood flow in the transferred LNs could be detected at 2 weeks after VLNT. At 6 months, the amount of reduced volume was 328.3 mL, the blood flow in the transferred LNs could be detected by ultrasonography, and the linear lymph flow from the recipient upper arm to the transferred LNs could be observed by ICG fluorescent lymphography (Figure 3B,C).

### 3.3. Mouse Model

The survival area of the skin paddle between the control, ischemic, and congestive groups was 99.2 ± 1.0%, 68.9 ± 7.7%, and 51.6 ± 8.9%, respectively (Figure 4). There were significant differences between the groups (*p* < 0.0001 in one-way ANOVA). There were significant differences between all groups, including the ischemic and congestive groups (*p* = 0.009 in Tukey–Kramer Method, Figure 5). In the survival area of the skin paddle, the vessel diameter was narrow, and erythrocytes were hardly observed in the vessel lumen in ischemic flaps. In contrast, in congestive flaps, the vessel diameter was increased, and the vessel lumen was filled with erythrocytes. Meanwhile, the leakage of erythrocytes into intercellular spaces was observed in the congestive group (Figure 6).

The internal structure of LNs in the control group was maintained (Figure 7). In ischemic LNs, HE staining showed an intravascular thrombus, and PTAH staining showed intravascular fibrin deposition in the medulla; however, the follicles retained their morphology, and there was no inflow of blood into the marginal sinus. In this group, a severe disruption of internal structures was not observed. In congestive LNs, HE staining showed a marked destruction of follicular structures. PTAH staining showed erythrocyte agglutination, and fibrin deposition was observed in vascular areas near the hilum and in the marginal sinus. Inflammatory cells were also observed around the clot in the marginal sinus.

Based on Myl9/12 immunostaining of follicles of LNs, only few Myl9/12-positive immunocompetent cells were observed in control LNs. In contrast, a small number were observed in ischemic LNs, while a large number were observed in congestive LNs.

In congestive LNs, the density of normal lymphocytes consisting of follicles decreased (Figure 8).

## 4. Discussion

This study showed that even in the absence of clear clinical signs of flap failure, poor clinical results of lymphedema treatment and low recanalization rates of lymphatic flow were observed in cases where blood flow in the transferred LNs could not be detected at 2 weeks after VLNT. These results suggest that the partial impairment of blood flow into the LNs after VLNT is one of the causes of poor outcomes. Herein, a mouse LN flap model was also developed for the first time to clearly distinguish partial ischemic flaps from congestive flaps and to observe the histological changes in LNs.

In clinical cases, partial blood flow impairment in the transferred LNs was observed by ultrasonography at 2 weeks after surgery in 11.9% of the cases despite having no abnormal findings in the anastomotic vessel monitoring for 72 h after surgery and no clinical symptoms for 2 weeks. In these cases, the amount of volume reduction was smaller than the others, and lymphatic vessel recanalization was not observed 6 months after surgery. As VLNT is a surgery performed to reconstruct the lymphatic function, the significantly reduced therapeutic effect observed in this study is critical. The loss of blood flow in LNs based on ultrasound findings should be regarded as treatment failure. In future studies, more detailed investigations of the timing and cause of blood flow disruption in LNs in the acute postoperative phase would be needed, and examining the effect of the microenvironment of VLNT on good or bad blood flow in the LN would be significant. The next issue was the histological changes in LNs when partial ischemia or congestion occurred in the flaps. As it was difficult to evaluate the transferred tissues histologically in clinical cases, a mouse model simulating VLNT was developed. With this model, the features of partial ischemic and congestive flaps could be observed. The area of necrosis was wider in the congestive flaps than in the ischemic flaps. In addition, intercellular hemorrhage was observed even in surviving areas in the congestive flap.

The histological changes in LNs included in the partial ischemic or congestive flaps were investigated. Both ischemia and congestion resulted in fibrin deposition associated with thrombus within blood vessels in LNs. Blood flow in LNs was compromised in both ischemia and congestion. In the congested LNs, erythrocytes, fibrin deposition, and surrounding inflammatory cells were also observed in the marginal sinus, which is the area where afferent lymphatics drain the lymph into. In contrast, no erythrocyte deposition and fibrin thrombosis were observed in the marginal sinus in the ischemic LNs. Physiologically, the lymphatic flow is slowed down in the marginal sinus of LNs from multiple afferent lymphatics to concentrate lymph [28,29]. These results suggest that the blood components, which leaked into the intercellular spaces in the skin and subcutaneous fat with blood vessel destruction, were collected in lymphatic capillaries and carried via the afferent lymphatic vessels to the marginal sinus of the LNs and formed clots in the congestive LN model. In congestive LN flaps, in addition to blood flow impairment in the LN itself, the disruption of the lymphatic inflow pathway may affect LN destruction and lead to an impaired lymphatic function.

Myl9/12 is a microparticle released from platelets upon activation [26,27]. Its release from platelets and the construction of Myl9/12 nets inside the blood vessels result in the release of inflammatory immunocompetent cells outside the blood vessels, leading to continued chronic inflammation [26,27]. In the newly developed mouse model, we observed numerous Myl9/12-positive immunocompetent cells, inflammatory cells around clots, and severe destruction of follicular structures only in congestive LNs.

Thus, a congested LN flap was thought to lose its function of transporting lymph due to an obstruction in the marginal sinus, along with blood flow congestion in LNs and subsequent inflammation. It is unclear whether this is directly related to a small skin paddle survival area in congestive flaps; however, the accumulation of blood in the stroma was observed in the congestive flap, which should induce a strong inflammatory response in soft tissues. In future studies, we would like to investigate the histological changes that occur in LNs in partial blood flow impairment flaps in the long term. In clinical cases, it was not possible to confirm whether the partial blood flow impairment in LN flaps was because of ischemia or congestion; however, the partial blood flow impairment in the flaps was considered to have a great influence on LN structures due to blood flow impairment in LNs themselves and blood clot formation in lymphatic pathways. In the acute phase after VLNT, it may be useful to routinely confirm the blood flow in the LNs and open the wound in case their blood flow has been partially impaired. In this study, we developed a model that closely resembled our case series, in which a part of the entire flap containing LNs exhibited arterial or venous insufficiency. There have also been reports of animal models in which only LNs were transferred. In future studies, we would like to compare the findings of ischemia and congestion in the model in which only LNs were transferred with the findings of our model [12,13,14,15,16].

The limitations of this study include the fact that in the clinical case series, no lymphatic recanalization was observed in 32.4% of patients even in Group G. This may be related to factors other than the blood flow in the transferred LNs. The axial consistency of lymphatic flow between the recipient and the flap or the presence or absence of intervening scarring between them are also important factors for lymphatic recanalization [30,31]. Moreover, the small number of cases in Group P was a limitation for the comparison between the two groups. However, poor blood flow was a factor for which a statistically significant difference was observed, although the number of cases was small. Further analysis of factors other than blood flow should be conducted in more cases in future studies.

Another limitation is the difference in various aspects between the lymphatic repair system of humans and mice. It is necessary to verify that the findings observed in mice do not always apply to humans.

## 5. Conclusions

The maintenance of blood flow in the transferred LNs following VLNT is an important factor influencing the therapeutic effect of VLNT. It is necessary to closely monitor partial blood flow impairment intraoperatively and postoperatively, especially in cases with congestion.

## Figures and Tables

**Figure 1 jcm-11-04052-f001:**
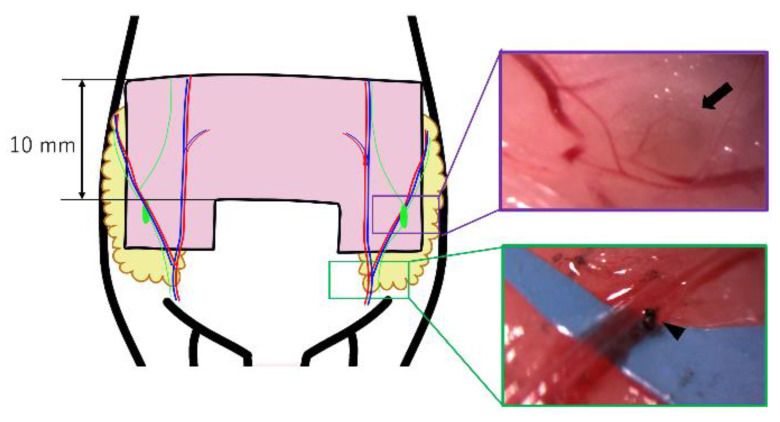
Flap design of the murine model of bilateral pedicled abdominal flaps with groin lymph nodes (LNs). (Purple square) Groin LNs and feeding vessels were identified and included in the flap. The black arrow indicates the left groin LN. (Green square) An artery or vein on the left side was ligated using a 10-0 nylon suture to produce an atrial ischemic or congestive flap. The black arrowhead indicates a venous ligation, which was 0.2 mm in width.

**Figure 2 jcm-11-04052-f002:**
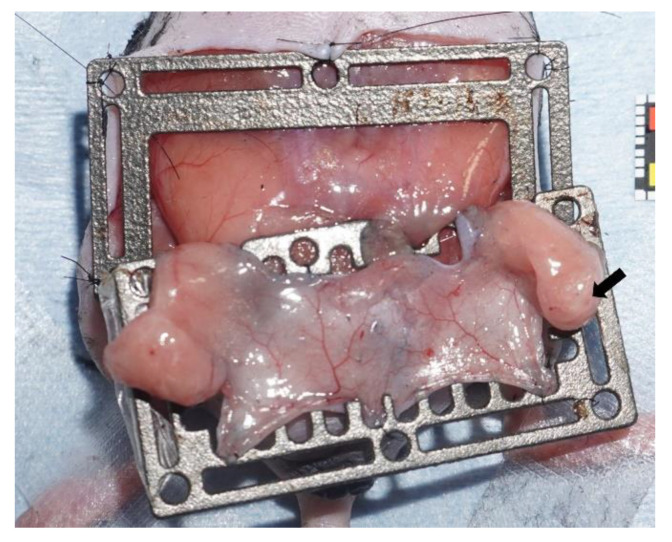
The flap was sutured to a metal frame, which was created using a 3D printer, to maintain a constant postoperative skin tension, prevent peripheral angiogenesis, and accurately measure the extent of the flap survival area for preventing angiogenesis from the surrounding skin.

**Figure 3 jcm-11-04052-f003:**
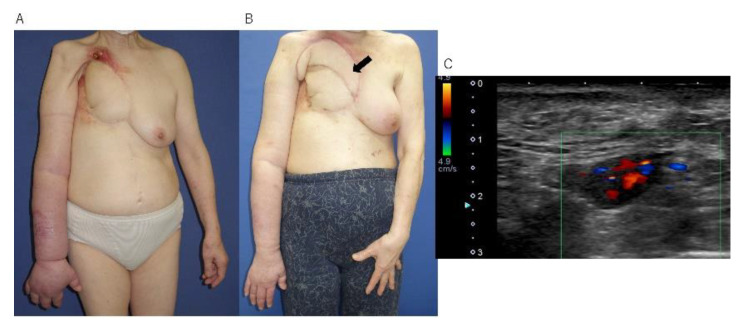
(**A**) A 72-year-old female patient suffered from right upper extremity lymphedema with a chronic wound on the thoracic wall after a mastectomy and radiotherapy. (**B**) After a vascularized lymph node transfer and chest wall reconstruction, the arm volume reduced by 275.0 m. The black arrow indicates the location of the transferred lymph nodes (LNs). (**C**) At 6 months after surgery, blood flow was clearly detected in the LNs.

**Figure 4 jcm-11-04052-f004:**
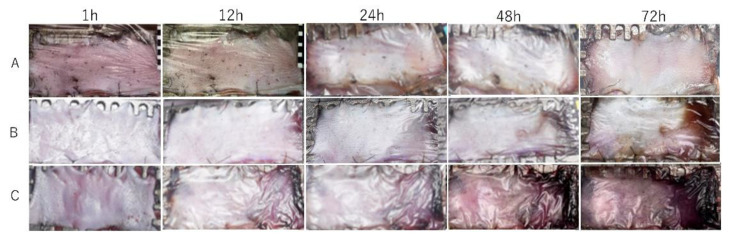
Typical color change patterns of skin paddles in the (**A**) control, (**B**) ischemia, and (**C**) congestive flaps until 72 h after surgery.

**Figure 5 jcm-11-04052-f005:**
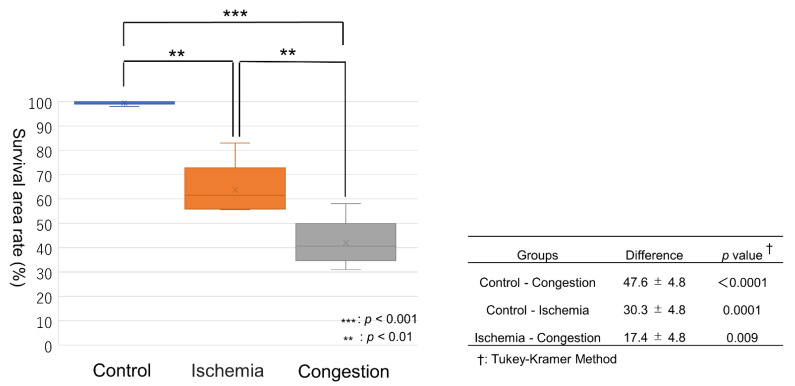
A difference in surviving skin paddle area rates was found between the control, ischemic, and congestive flaps.

**Figure 6 jcm-11-04052-f006:**
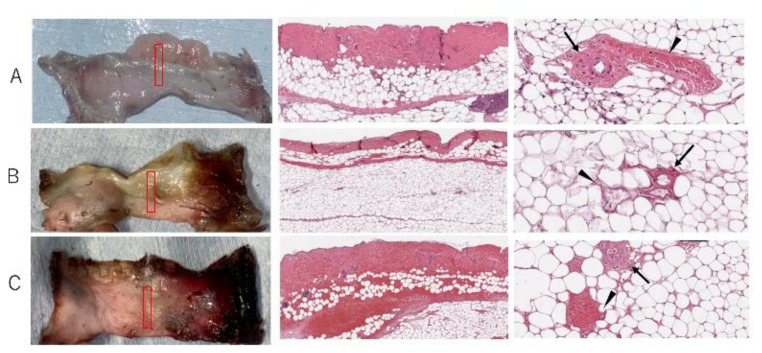
Histological findings in surviving skin paddle areas in the (**A**) control, (**B**) ischemic, and (**C**) congestive flaps at 72 h. The black arrow indicates an artery, whereas the black arrowhead indicates a vein. In the congestive flaps, blood cells leaked into intercellular spaces, and the vessel lumen was filled with blood.

**Figure 7 jcm-11-04052-f007:**
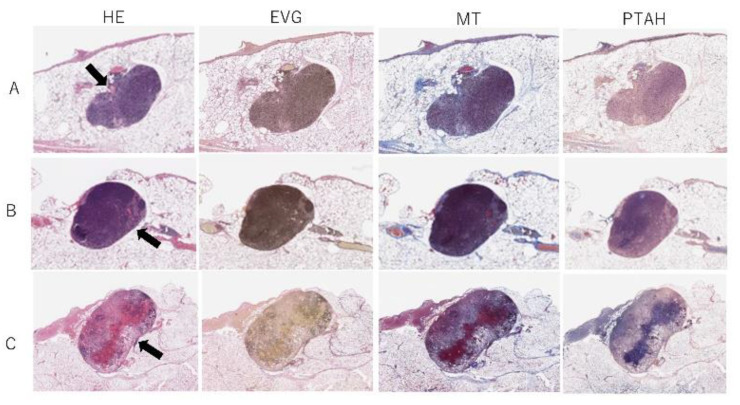
Histological findings of (**A**) control, (**B**) ischemic, and (**C**) congestive lymph nodes (LNs). Black arrows in hematoxylin and eosin stain indicate hilar sites: (**A**) The control LNs maintained their internal structure. (**B**) In ischemic LNs, intravascular thrombus and intravascular fibrin deposition were observed in the medulla. (**C**) In congestive LNs, erythrocyte agglutination and fibrin deposition were observed in vascular areas near the hilum and in the marginal sinus. Moreover, inflammatory cells were observed around the clot in the marginal sinus. HE, hematoxylin and eosin; EVG, Elastica van Gieson; MT, Masson’s trichrome; PTAH, Phosphotungstic acid hematoxylin.

**Figure 8 jcm-11-04052-f008:**
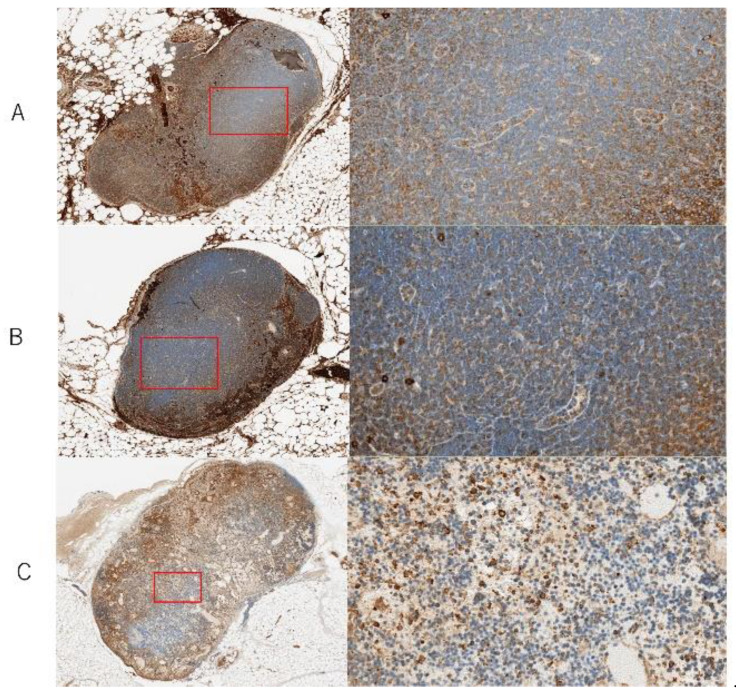
Myosin light chain 9, 12a, and 12b (Myl9/12) immunostaining of follicles at 72 h after surgery. The left image shows the entire LN and a magnified image of the red square area on the right (×400). (**A**) In control LNs, very few Myl9/12-positive immunocompetent cells were observed. (**B**) In ischemic LNs, small numbers of Myl9/12-positive immunocompetent cells were observed. (**C**) In congestive LNs, numerous Myl9/12-positive immunocompetent cells were observed.

**Table 1 jcm-11-04052-t001:** Patient characteristics and study results. Recanalization was confirmed by indocyanine green fluorescent lymphography.

Group	Good Blood Flow in LN	Poor Blood Flow in LN	*p*-Value
Number of patients	37	5	
Age (years)	55.4 ± 8.8	57.0 ± 7.2	0.86
BMI (kg/m^2^)	22.9 ± 3.1	24.6 ± 3.4	0.24
Surgical method	DIEP + VLNT (SCIA): 18 VLNT (SCIA): 12 VLNT (TDA): 7	DIEP + VLNT (SCIA): 3 VLNT (SCIA): 1 VLNT (TDA): 1	0.17
Volume improvement (mL)	142.9 ± 89.4	62.1 ± 55.0	0.03
Recanalization	25 (67.6%)	0 (0%)	0.007

BMI, body mass index; LN, lymph node; DIEP, breast reconstruction using deep inferior epigastric artery perforator flap; VLNT, vascularized lymph node transfer; SCIA, superficial circumflex iliac artery; TDA, thoracodorsal dorsal artery.

## Data Availability

The data presented in this study are available on request from the corresponding author.

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
