# Peer review of "Clinical and Histological Effects of Partial Blood Flow Impairment in Vascularized Lymph Node Transfer"

_jcm, 2022, doi:10.3390/jcm11144052_

Round 1

Reviewer 1 Report

Summary: The authors studied the effect of partial blood flow impairment of vascularized lymph node transfer in lymphedema patients through a clinical study and animal model. This study found that partial blood flow impairment in lymph nodes will decrease the improvement following VLNT in lymphedema patients. LN with good blood flow improved the limb volume ~2 times more than LN with poor blow flow. Only LN with good blood flow showed 67.6% of recanalization after the VLNT. This study is interesting however, some of the concerns should be addressed and overall writing and data presentation should be improved.

1.       There are 8 figures but only 3 of them were addressed in the manuscript. The entire animal experiments were not described at all. All of the data presented in the paper have to be explained in the manuscript.

2.       The size of the poor blood flow group is only 5 and that is too small compared to the good flow group.

3.       There is a concern about this animal model since it does not seem to show the effect of “good blood flow in transferred LN”. The animal model used here could show the effect of the flow in the skin and pre-existing lymph nodes, but it does not necessarily represent good/bad blood flow of transferred lymph nodes. There are other models such as popliteal lymph node transfer following popliteal lymph node dissection-induced lymphedema, which will be more related to the clinical study in this manuscript. Local injection of VEGFR2 inhibitor will reduce the vascularization in the transferred lymph node.

4.       In the clinical study, what is the effect of the microenvironment of VLNT on good/bad blood flow in lymph nodes? Do you see any difference in VEGFA level in the patient’s tissue or serum between group G and P?

5.       Is there any difference in inflammatory cytokine levels in serum of group G and P?

6.       Table1, patients with good blood flow in LN should be 37?

7.       What does it mean ‘volume improvement’? Does it mean reduced limb volume after VLNT? If it is, it should be clarified in the method.

8.       References are not marked correctly. For example, page3 line 118 has to be [19] not [20]

Author Response

Thank you very much for all the appropriate and constructive advice.

I have made responses to each of your suggestions.

Thank you very much.

Comments and Suggestions for Authors

Summary: The authors studied the effect of partial blood flow impairment of vascularized lymph node transfer in lymphedema patients through a clinical study and animal model. This study found that partial blood flow impairment in lymph nodes will decrease the improvement following VLNT in lymphedema patients. LN with good blood flow improved the limb volume ~2 times more than LN with poor blow flow. Only LN with good blood flow showed 67.6% of recanalization after the VLNT. This study is interesting however, some of the concerns should be addressed and overall writing and data presentation should be improved.

  1. There are 8 figures but only 3 of them were addressed in the manuscript. The entire animal experiments were not described at all. All of the data presented in the paper have to be explained in the manuscript.

Response 1:

Thank you very much for pointing it out. As you pointed out, the necessary parts were omitted due to an error. We added the description of Figures 4 -8 in the manuscript as follows;

3.3. Mouse model

The survival area of the skin paddle between the control, ischemic, and congestive groups were 99.2% ± 1.0%, 68.9% ± 7.7%, and 51.6%± 8.9%, respectively (Figure 4). There were significant differences between the groups (P < 0.0001 in one-way ANOVA). There were significant differences between all groups, including the ischemic and congestive groups (p = 0.009 in Tukey–Kramer Method, Figure 5). In the survival area of the skin paddle, the vessel diameter was narrow, and erythrocytes were hardly observed in the vessel lumen in ischemic flaps. In contrast, in congestive flaps, the vessel diameter was increased, and the vessel lumen was filled with erythrocytes. Meanwhile, the leakage of erythrocytes into intercellular spaces was observed in the congestive group (Figure 6).

The internal structure of LNs in the control group was maintained (Figure 7). In ischemic LNs, HE staining showed intravascular thrombus, and PTAH staining showed intravascular fibrin deposition in the medulla; however, the follicles retained their morphology, and there was no inflow of blood into the marginal sinus. In this group, a severe disruption of internal structures was not observed. In congestive LNs, HE staining showed marked destruction of follicular structures. PTAH staining showed erythrocyte agglutination, and fibrin deposition was observed in vascular areas near the hilum and in the marginal sinus. Inflammatory cells were also observed around the clot in the marginal sinus.

Based on myl9/12 immunostaining of follicles of LNs, only few myl9/12-positive immunocompetent cells were observed in control LNs. In contrast, a small number was observed in ischemic LNs, while a large number was observed in congestive LNs.

In congestive LNs, the density of normal lymphocytes consisting of follicles decreased (Figure 8).

  1. The size of the poor blood flow group is only 5 and that is too small compared to the good flow group.

Response 2:

As you pointed out, the large difference in the number of cases between the two groups should be noted in the limitation. The number of patients in each group is estimated to be the same even if the number of patients is increased in the future because the operation is performed with consideration for blood flow in the LNs. On the other hand, there was a clear difference in treatment outcome that was statistically significant despite such a difference in the number of cases. We have revised the text as follows;

Moreover, the small number of cases in Group P was a limitation for the comparison between the two groups. However, poor blood flow was a factor for which a statistically significant difference was observed, although the number of cases was small.

  1. There is a concern about this animal model since it does not seem to show the effect of “good blood flow in transferred LN”. The animal model used here could show the effect of the flow in the skin and pre-existing lymph nodes, but it does not necessarily represent good/bad blood flow of transferred lymph nodes. There are other models such as popliteal lymph node transfer following popliteal lymph node dissection-induced lymphedema, which will be more related to the clinical study in this manuscript. Local injection of VEGFR2 inhibitor will reduce the vascularization in the transferred lymph node.

Response 3

Thank you very much for your advice. In this study, we developed a model that closely resembled our case series, in which a part of the entire flap containing LN exhibited arterial or venous insufficiency. In the revised version, the intent of the animal model of this study was specified, and in the Introduction and Discussion, we described about the other animal models. The discussion noted that these animal models should be compared with the animal models in this study in the future to verify the differences. In the Revised version, we added the following sentences;

Introduction

In animal models of vascularized, single-venous, and non-vascularized LN transfers, survival rates as well as lymphatic recanalization with the recipient area have been reported [12-16].

Discussion

In this study, we developed a model that closely resembled our case series, in which a part of the entire flap containing LN exhibited arterial or venous insufficiency. There have also been reports of animal models in which only LNs were transferred. In future studies, we would like to compare the findings of ischemia and congestion in the model in which only LNs were transferred with the findings of our model [12-16].

  1. Tinhofer IE, Yang CY, Chen C, Cheng MH. Impacts of arterial ischemia or venous occlusion on vascularized groin lymph nodes in a rat model. J Surg Oncol. 2020;121(1):153-62.
  2. Cornelissen AJ, Qiu SS, Lopez Penha T, Keuter X, Piatkowski de Grzymala A, Tuinder S, et al. Outcomes of vascularized versus non-vascularized lymph node transplant in animal models for lymphedema. Review of the literature. J Surg Oncol. 2017;115(1):32-6.
  3. Tobbia D, Semple J, Baker A, Dumont D, Johnston M. Experimental assessment of autologous lymph node transplantation as treatment of postsurgical lymphedema. Plast Reconstr Surg. 2009;124(3):777-86.
  4. Visconti G, Constantinescu T, Chen PY, Salgarello M, Franceschini G, Masetti R, et al. The Venous Lymph Node Flap: Concepts, Experimental Evidence, and Potential Clinical Implications. J Reconstr Microsurg. 2016;32(8):625-31.
  5. Ishikawa K, Maeda T, Funayama E, Hayashi T, Murao N, Osawa M, et al. Feasibility of pedicled vascularized inguinal lymph node transfer in a mouse model: A preliminary study. Microsurgery. 2019;39(3):247-54.

  1. In the clinical study, what is the effect of the microenvironment of VLNT on good/bad blood flow in lymph nodes? Do you see any difference in VEGFA level in the patient’s tissue or serum between group G and P?

Response 4

Thank you for pointing it out. The VEGF family was not investigated in this study. Based on the results of this research, the following description has been added in order to investigate the points you pointed out in future studies;

In future studies, more detailed investigations of the timing and cause of blood flow disruption in LNs in the acute postoperative phase would be needed, and examining the effect of the microenvironment of VLNT on good or bad blood flow in the LN would be significant.

  1. Is there any difference in inflammatory cytokine levels in the serum of group G and P?

Response 5

In this study, we did not examine blood samples at a uniform timing. Based on the results of this research, the following description has been added in order to investigate the points you pointed out in future research;

In future studies, more detailed investigations of the timing and cause of blood flow disruption in LNs in the acute postoperative phase would be needed, and examining the effect of the microenvironment of VLNT on good or bad blood flow in the LN would be significant.

  1. Table1, patients with good blood flow in LN should be 37?

Response 6

There was a mistake in the description. We corrected it.

  1. What does it mean ‘volume improvement’? Does it mean reduced limb volume after VLNT? If it is, it should be clarified in the method.

Response 7

We have stopped using the term “Volume improvement” and have clearly defined it in methods as the amount of volume reduction achieved by treatment as follows;

The limb volume was calculated using the circumference of the four points and the distance between the measurement sites [17]. The difference between the preoperative and postoperative limb volumes was defined as the amount of volume reduction achieved by the treatment.

  1. References are not marked correctly. For example, page3 line 118 has to be [19] not [20]

Response 8

I'm sorry. The reference and its contents have been reviewed and corrected, including some references added in the revised version.

Reviewer 2 Report

This study aimed to determine the clinical and histological impacts of partial blood flow impairment to the transferred LNs after VLNT. Meanwhile, authors would like to explore whether insufficient arterial or venous blood flow is the more serious problem. The concept is novel. The author’s work is well done. While, some questions need to be considered and explained:

1.       In the part of Introduction, there are too many citations in the first sentence.

2.       Some scholars have proved in animal experiments that both single venous lymph nodes and lymph nodes without blood supply can survive after transplantation, and can form lymphatic recanalization with the recipient area. Please add relevant literatures into the introduction.

3.       The definition of ‘partially impaired blood flow’ needs to be supplemented in the introduction. Does it refer to single impaired arterial supply, or single impaired venous supply, or both? What’s different between ‘partially impaired blood flow’ and ‘impaired blood flow’?

4.       In the part of Methods, please clarify how to realize ‘immediately after the operation, compression therapy for lymphedema was restarted, but compressing the vascular anastomosis sites was avoided’.

5.       Why do the authors choose 2 weeks after surgery as the time point to differentiate Group G and Group P?

6.       It is necessary to clarify in Methods whether the sampling time of ischemic and congestive LNs is 36 hours after surgery. Why is there no longer observation, for example, 1 week, 2 weeks, 1 month after surgery? It is recommended to supplement longer histological observations of these mice groups.

7.       Lack of statistical methods introduction in the part Methods.

8.       The full names of HE, EVG, MT, and PTAH should be indicated in the result figure 7. The findings of groups A, B, and C should be marked in the corresponding pictures.

9.       I can’t get any useful information from the Supplementary Materials.

Author Response

Thank you very much for all the appropriate and constructive advice.

I have made responses to each of your suggestions.

Thank you very much.

This study aimed to determine the clinical and histological impacts of partial blood flow impairment on the transferred LNs after VLNT. Meanwhile, the authors would like to explore whether insufficient arterial or venous blood flow is the more serious problem. The concept is novel. The author’s work is well done. While some questions need to be considered and explained:

  1. In the part of Introduction, there are too many citations in the first sentence.

Response 1

We reduced the number of citations in the first sentence to only the appropriate studies.

  1. Some scholars have proved in animal experiments that both single venous lymph nodes and lymph nodes without blood supply can survive after transplantation, and can form lymphatic recanalization with the recipient area. Please add relevant literatures into the introduction.

Response 2

We have changed our understanding that it is very important to fully consider your points. In the Introduction, we have added papers from several studies related to the animal models you mentioned. In the Discussion, we added the description about the importance of comparing and verifying the models used in these papers and the models used in this study in future studies. The added sentences are as follows;

Introduction

In animal models of vascularized, single-venous, and non-vascularized LN transfers, survival rates as well as lymphatic recanalization with the recipient area have been reported [12-16].

Discussion

In this study, we developed a model that closely resembled our case series, in which a part of the entire flap containing LN exhibited arterial or venous insufficiency. There have also been reports of animal models in which only LNs were transferred. In future studies, we would like to compare the findings of ischemia and congestion in the model in which only LNs were transferred with the findings of our model [12-16].

  1. Tinhofer IE, Yang CY, Chen C, Cheng MH. Impacts of arterial ischemia or venous occlusion on vascularized groin lymph nodes in a rat model. J Surg Oncol. 2020;121(1):153-62.
  2. Cornelissen AJ, Qiu SS, Lopez Penha T, Keuter X, Piatkowski de Grzymala A, Tuinder S, et al. Outcomes of vascularized versus non-vascularized lymph node transplant in animal models for lymphedema. Review of the literature. J Surg Oncol. 2017;115(1):32-6.
  3. Tobbia D, Semple J, Baker A, Dumont D, Johnston M. Experimental assessment of autologous lymph node transplantation as treatment of postsurgical lymphedema. Plast Reconstr Surg. 2009;124(3):777-86.
  4. Visconti G, Constantinescu T, Chen PY, Salgarello M, Franceschini G, Masetti R, et al. The Venous Lymph Node Flap: Concepts, Experimental Evidence, and Potential Clinical Implications. J Reconstr Microsurg. 2016;32(8):625-31.
  5. Ishikawa K, Maeda T, Funayama E, Hayashi T, Murao N, Osawa M, et al. Feasibility of pedicled vascularized inguinal lymph node transfer in a mouse model: A preliminary study. Microsurgery. 2019;39(3):247-54.

  1. The definition of ‘partially impaired blood flow’ needs to be supplemented in the introduction. Does it refer to single impaired arterial supply, or single impaired venous supply, or both? What’s different between ‘partially impaired blood flow’ and ‘impaired blood flow’?

Response 3

In the original version, the definition of “partially impaired blood flow ” was not explained, which could confuse journal readers. The definition was accurately described in the Introduction. Also, since partial impaired blood flow and impaired blood flow were used interchangeably, I think that this could have confused journal readers. We have unified all descriptions to "partial impaired blood flow".

As there are cases with insufficient blood flow in the transferred LNs after surgery despite the skin paddle survival and no clinical complications in the acute phase, we emphasize the importance of postoperative ultrasound monitoring of the transplanted lymph nodes (LNs) to confirm the maintenance of blood flow and bilayer structure [11]. In this study, partial blood impairment, regardless of arterial blood flow, venous blood flow, or both arterial and venous blood flows, was defined as the condition in which the blood flow of the transferred LN was not confirmed even though the skin paddle survived.

  1. In the part of Methods, please clarify how to realize ‘immediately after the operation, compression therapy for lymphedema was restarted, but compressing the vascular anastomosis sites was avoided’.

Response 4

In the Methods of the revised manuscript, specific methods for avoiding pressure on vascular anastomosis sites are described as follows;

Compression therapy for lymphedema was restarted immediately after the surgery. The vascular anastomosis site was marked with a pen on the skin and was covered with a clear film, and the compression of the site was prohibited for the first 10 days.

  1. Why do the authors choose 2 weeks after surgery as the time point to differentiate Group G and Group P?

Response 5

The reason why the time point of 2 weeks after surgery was used is that in the past reports of breast reconstruction by free flap, there were reports of total necrosis until 12 days after surgery, and by using the time point of 2 weeks after surgery, the effect of total necrosis on the results could be eliminated. In addition to the relevant references and references, we have added the following to Methods;

At 2 weeks after surgery, when the risk of total flap failure was considered minimal, ultrasonography of the transferred LNs was performed to determine whether it was possible to examine their blood flow [22,23].

  1. Zoccali G, Molina A, Farhadi J. Is long-term post-operative monitoring of microsurgical flaps still necessary? J Plast Reconstr Aesthet Surg. 2017;70(8):996-1000.
  2. Nelson JA, Kim EM, Eftakhari K, Low DW, Kovach SJ, Wu LC, et al. Late venous thrombosis in free flap breast reconstruction: strategies for salvage after this real entity. Plast Reconstr Surg. 2012;129(1):8e-15e.

  1. It is necessary to clarify in Methods whether the sampling time of ischemic and congestive LNs is 36 hours after surgery. Why is there no longer observation, for example, 1 week, 2 weeks, 1 month after surgery? It is recommended to supplement longer histological observations of these mice groups.

Response 6

In the revised manuscript, it was noted in Methods that the sampling time was 72 h. The purpose of this study using the animal model was to observe the histological findings of 72 h in order to observe what is happening in the LN in the period in which the LN was not depicted by the ultrasonic examination because the biopsy was practically impossible in clinical cases. How lymph nodes degenerate pathologically in long-term observation is a very interesting topic and should be addressed in future studies. I would like to address this issue as a separate research theme in the future, while thoroughly examining the risk of infection and death in mice. The following description has been added to the revised version.

Methods

Histopathological specimens of LNs and surrounding tissues were obtained 72 h after flap elevation.

Discussion

In future studies, we would like to investigate the histological changes that occur in LNs in partial blood flow impairment flaps in the long term.

  1. Lack of statistical methods introduction in the part Methods.

Response 7

Statistical methods were added to the Methods as follows;

2.3. Statistical analysis

JMP version 13 software (SAS Institute Inc., Cary, NC, USA) was used for all statistical analyses. During comparisons between the two groups in a clinical study, the t-test was used to compare continuous variables, and the chi-square test was used to compare frequencies. During the comparison of continuous variables among the three groups in animal study, multiple tests were performed; the one-way analysis of variance (ANOVA) and the Tukey–Kramer method were used. A p-value of <0.05 was considered statistically significant.

  1. The full names of HE, EVG, MT, and PTAH should be indicated in the result figure 7. The findings of groups A, B, and C should be marked in the corresponding pictures.

Response 8

Thank you for pointing it out. It was inappropriate not to mention what you pointed out. The Revised version has added descriptions about them.

  1. I can’t get any useful information from the Supplementary Materials.

Response 9

The supplementary video has been deleted.

Round 2

Reviewer 1 Report

The authors significantly improved the manuscript, however, I still do have a concern about the animal model.

Reviewer 2 Report

The revised version is better.